# Structural and Viscoelastic Properties of Thermoplastic Polyurethanes Containing Mixed Soft Segments with Potential Application as Pressure Sensitive Adhesives

**DOI:** 10.3390/polym13183097

**Published:** 2021-09-14

**Authors:** Mónica Fuensanta, José Miguel Martín-Martínez

**Affiliations:** Adhesion and Adhesives Laboratory, University of Alicante, 03080 Alicante, Spain; monica.fuensanta@ua.es

**Keywords:** thermoplastic polyurethane, mixed soft segments, pressure sensitive adhesive, structure-property relationship, viscoelastic properties

## Abstract

Thermoplastic polyurethanes (TPUs) were synthetized with blends of poly(propylene glycol) (PPG) and poly(1,4-butylene adipate) (PAd) polyols, diphenylmethane-4,4′-diisocyanate (MDI) and 1,4-butanediol (BD) chain extender; different NCO/OH ratios were used. The structure and viscoelastic properties of the TPUs were assessed by infrared spectroscopy, differential scanning calorimetry, X-ray diffraction, thermal gravimetric analysis and plate-plate rheology, and their pressure sensitive adhesion properties were assessed by probe tack and 180° peel tests. The incompatibility of the PPG and PAd soft segments and the segregation of the hard and soft segments determined the phase separation and the viscoelastic properties of the TPUs. On the other hand, the increase of the NCO/OH ratio improved the miscibility of the PPG and PAd soft segments and decreased the extent of phase separation. The temperatures of the cool crystallization and melting were lower and their enthalpies were higher in the TPU made with NCO/OH ratio of 1.20. The moduli of the TPUs increased by increasing the NCO/OH ratio, and the tack was higher by decreasing the NCO/OH ratio. In general, a good agreement between the predicted and experimental tack and 180° peel strength values was obtained, and the TPUs synthesized with PPG+PAd soft segments had potential application as pressure sensitive adhesives (PSAs).

## 1. Introduction

Thermoplastic polyurethanes (TPUs) are versatile polymers commonly used as elastomers, adhesives and coatings in automotive, building and construction, shoe manufacturing and biomedical technology, among others. The structure of the TPUs is tightly related to their properties (structure-property relationship) and is determined by the nature of the reactants, the reaction conditions, etc [1,2]. TPU structure consists of soft (SS) and hard (HS) segments, the SS are due to the interactions between the long polyol chains, and the HS are produced by reacting the diisocyanate with the polyol and the chain extender (short chain diol) [1,2]. Polyether, polyester, polycaprolactone and polycarbonate diols with molecular weights of 600–3000 g/mol are the most common polyols in the synthesis of the TPUs [3], and their properties are noticeably influenced by the polyol’s nature [4]. Thus, the TPUs made with polyether polyols show good hydrolysis resistance, flexibility and water-vapor permeability, but poor mechanical properties, and the TPUs made with polyester polyols combine higher tensile strength and hardness and good resistance to oxidation, but poor hydrolytic resistance [5,6]. Several previous studies have considered the influence of the nature, molecular weight and structure of the SS made with one polyol on the polyurethane’s properties [7,8,9,10,11]. The miscibility of the SS provided different degrees of micro-phase separation and distinct hydrogen bonds interactions among the HS [1,2,12,13,14,15,16,17,18]. However, the structure-property relationship of the TPUs synthesized with two polyols of a different nature, i.e., TPUs with two kinds of SS, have been less studied [19,20,21,22,23,24,25].

It has been shown elsewhere [19,21] that the existence of two kinds of soft segments of different nature and composition allowed the control of the micro-phase separation of the polyurethanes for designing their mechanical, thermal and surface properties. In fact, similar mechanical-optical properties were obtained in thin urethane/urea polymers made with polybutadiene and propylene oxide mixed soft segments than in liquid crystalline polymers intended as mechanically activated sensors [21]. Mirhosseini et al. [24] have shown that the use of mixed soft segments accelerated the micro-phase separation process as compared with TPUs containing one type of polyol, and the TPUs made with mixed soft segments showed enhanced elongation-at-break and toughness. In the same direction, Cui et al. [25] have shown that the introduction of polycaprolactone (PCL) polyol promoted micro-phase mixing and crystallization, and reduced micro-phase separation in the polyurethanes better than polytetramethyleneglycol polyol did; this was ascribed to the stronger carbonyl group in PCL in forming hydrogen bonds with —NH bond. On the other hand, the polyurethanes synthetized with MDI, 1,4-butanediol chain extender and polyester polyols of different nature and molecular weights showed different degrees of crystallinity and mobility of the HS depending on the length of the carbon atoms in the polyols [19].

Most studies dealing with polyurethanes synthesized with two kinds of soft segments have been devoted to the degree of micro-phase separation induced by their different miscibility. Some interesting micro-phase morphologies have been found in TPUs with 25–50% HS synthesized by reacting MDI and two chemically different soft segments—poly(tetramethylene oxide) (PTMO) or poly(hexylethyl carbonate) (PHEC) and polydimethylsiloxane (PDMS) or poly-isobutylene (PIB)—with molecular weights of 500–5000 g/mol [19], stronger hydrogen bond interactions between the urethane HS than the ones between the urethane HS and the SS segments were obtained; furthermore, the presence of the non-polar soft segments favoured the micro-phase separation in the TPUs [19]. Similarly, the urethane/urea polymers synthesized with two highly dispersed poly (propylene oxide)—PPG—and polybutadiene SS showed weaker hydrogen bonds among the HS by increasing the polybutadiene content due to higher mobility of the HS [20]. Furthermore, the miscibility and micro-phase separation of the polyurethanes synthesized with 1,6-hexamethylene diisocyanate (HDI), PPG polyols with molecular weights of 400–4000 g/mol, and poly (ethylene oxide) (PEO) with molecular weight of 600 g/mol, were ascribed to the incompatibility of the soft segments and was not induced by hard/soft segment segregation [22]. In addition, the micro-phase separation of the TPUs synthesized with PPG and hydroxyl-terminated polybutadiene—HTPB—polyols consisted of a triple-phase structure of hard domains, HTPB rich phase and mesophase, this mesophase consisted of hard domains, PPG and HTPB segments excluded out of the urethane and HPTB rich domains [23].

None of the previous studies have considered the viscoelastic and adhesion properties of TPUs synthesized with two kinds of SS. These properties are critical in adhesives, mainly in pressure sensitive adhesives (PSAs) which are the main interest of this study.

PSAs are used in labels, tapes, note pads, medical tapes and bandages, among other applications, and they are characterized by non-permanent reversible adhesion properties [26]. The performance of the PSAs is tightly related to their rheological and viscoelastic properties as they have to be mainly elastic during de-bonding and mainly viscous during bonding to a given substrate [27,28,29]. An adequate PSA must have the right balance between adhesion (peel, tack) and cohesion properties, and this is difficult to reach because of the need for a balanced elastic-viscous behaviour [30]. One way of balancing the elastic and viscous properties in PSAs is the use of segmented polymers; thus, TPUs are promising raw materials for PSAs.

Several studies dealing with waterborne, hybrid acrylic-urethane, polysiloxane-urethane and thermoplastic based polyurethane PSAs have been considered in the existing literature [31,32,33,34,35,36,37,38,39,40,41]. The tack is not a characteristic property of the polyurethane PSAs, and the addition of tackifiers has been proposed [42]. The pressure sensitive adhesive properties of the polyurethanes have been imparted by using polyols with one reactive OH group of low molecular weight and low NCO/OH ratios during their synthesis [43]. Several strategies have been proposed for improving the adhesion of the polyurethane PSAs, such as the addition of trimethylolpropane/toluene-2,4-diisocyanate cross-linker [31], the use of different hexamethylene di-isocyanate trimer ratios [44] and the crosslinking with electron beam radiation [45].

Several thermoplastic polyurethane-based PSAs (TPU PSAs) made with mixtures of polyols of different nature and molecular weights have also been proposed. Akram et al. [32] synthetized grafted PU PSAs with mixtures of polypropylene glycols (PPGs) of different molecular weights and hydroxyl-terminated polybutadiene (HTPB), and the increase of the polybutadiene content increased the tack. Huang et al. [46] have synthesized prepolymers with toluene di-isocyanate and mixture of castor oil and polypropylene glycol polyols cured at different temperatures, and they showed adequate peel strength. On the other hand, the performance of TPU PSAs made with MDI and polyether polyols of different nature and molecular weights has been shown, in which the role of the degree of micro-phase separation on their performance has been demonstrated [37,38]. These TPU PSAs showed good tack at 10–37 °C and their structures were controlled by the soft domains with minor contribution of the bonded urethane species and an important degree of micro-phase separation [37]. In another study, the TPU PSAs synthesized with PPG polyols with molecular weights of 450 and 2000 g/mol, an NCO/OH ratio of 1.1 and 12.5–38.7% HS content have shown different performance; the increase of the HS content increased the percentage of the hydrogen bonded urethane species and produced a lower degree of micro-phase separation. The TPUs with HS content lower than 27.9% showed high tack and adequate debonding properties [38]. Furthermore, the TPU PSAs synthesized with different blends of PPG and poly(tetramethylene ether glycol) (PTMEG) polyether diols and different NCO/OH ratios had semicrystalline regions in the SS which inhibited the mobility of the polymeric chains. The increase of the PTMEG content in the polyols blends improved both the cohesion and the adhesion but decreased the tack of the TPU PSAs [40]. In addition, TPUs with 13.9–24.4% HS contents have been synthesized with different blends of PPG of molecular weight 2000 g/mol and poly(tetrahydrofuran) (PTHF) of molecular weight 1000 g/mol, and the increase of the HS content increased the contribution of the associated by hydrogen bond urethane species and decreased the degree of micro-phase separation due to the enhanced interactions between the HS; this led to improved cohesion [41]. Finally, PSAs made with blends of TPUs with satisfactory tack, cohesion and adhesion properties have been developed, their improvements derived from the higher number of hydrogen bonds among the HS, which led to lower degree of micro-phase separation than in the parent TPUs [39].

In previous studies [32,37,38,39,40,41,46], an adequate balance between adhesion and cohesion in TPU PSAs made with blends of polyols required a relatively important HS content or chemical cross-linking. The novelty of this study consists of the synthesis of TPUs with blends of polyether and polyester polyols and low HS content (13–15%) without any chemical cross-linker for obtaining TPU PSAs, an approach which differs substantially from the existing literature. The miscibility of the SS should determine the structure, degree of micro-phase separation and viscoelastic properties of the TPUs, and, therefore, their performance as PSAs. In fact, the polyester SS show both dispersive and dipole-dipole van der Waals interactions, whereas the polyether SS show dispersive van der Waals interactions only; as such, it can be anticipated that the miscibility of the mixed polyether and polyester SS in the TPUs will be different. The structure, viscoelastic and adhesion properties of the TPUs and the TPU PSAs synthesized with blends of polyether and polyester polyols have been carried out while paying particular focus to the role of the miscibility of the SS on their viscoelastic and adhesion properties.

## 2. Materials and Methods

### 2.1. Materials

Diphenylmethane-4,4′-diisocyanate (MDI) flakes with 33.6 wt.% NCO (Desmodur^®^ 44 MC—Covestro, Leverkusen, Germany) were used without further purification. Poly(propylene glycol) (PPG) with molecular weight of 2000 g/mol and hydroxyl number of 56 mg KOH/g (Alcupol^®^ 2021—Repsol, Madrid, Spain) and poly(1,4-butylene adipate) (Pad) with molecular weight of 2000 g/mol and hydroxyl number of 54 mg KOH/g (Hoopol F501—Synthesia Technology, Barcelona, Spain) polyols were used. They were dried at 80 °C under 300 mbar for 2 h before use. The chain extender, 1,4-butanediol (BD) and the tin catalyst, dibutyltin dilaurate (DBTDL) were supplied by Sigma Aldrich (St. Louis, MO, USA).

### 2.2. Synthesis of the Thermoplastic Polyurethanes

Different TPUs were synthetized by using the prepolymer method (Figure 1) with single or mixed PPG and PAd soft segment (SS) and very similar hard segment (HS) content of 13–15 wt%; NCO/OH ratios of 1.13, 1.20 and 1.30 were used. MDI was melted at 70 °C in a 500 cm^3^ four-necked flask under nitrogen atmosphere, and PPG, PAd or PPG+PAd mixtures were added under stirring at 250 rpm with a stainless steel rod coupled to a Heidolph overhead stirrer RZR-2000 (Kelheim, Germany). After 30 min, the stirring was decreased to 80 rpm and 0.028 g catalyst was added, allowing the reaction to occur at 80 °C and 80 rpm for 2 h. The linear NCO-ended prepolymer was fully reacted with 1,4-butanediol (BD) chain extender at 80 °C and 80 rpm for 10–15 min for obtaining the segmented TPUs. The nomenclatures and the amounts of reactants used in the synthesis of the TPUs are summarized in Table 1.

### 2.3. Experimental Techniques

#### 2.3.1. Attenuated Total Reflectance-Fourier Transform Infrared (ATR-IR) Spectroscopy

The chemical structure and degree of micro-phase separation of the TPUs were analyzed by ATR-IR spectroscopy. The ATR-IR spectra was collected in a Tensor 27 FT-IR spectrometer (Bruker Optik GmbH, Ettlinger, Germany) by using a Golden Gate single reflection diamond ATR accessory. All ATR-IR spectra were normalized to the most intense absorbance band. On the other hand, the curve fitting of the carbonyl region (1800–1650 cm^−1^) of the ATR-IR spectra was performed with the Origin 8.0 software program; the Gaussian function provided the best fit.

#### 2.3.2. Differential Scanning Calorimetry (DSC)

The structure and degree of micro-phase separation of the TPUs were analysed in a DSC Q100 equipment (TA instruments, New Castle, USA). TPU samples (8–9 mg) were placed in a hermetically closed aluminum pan. Three consecutive cycles from −80 to 250 °C at 10 °C/min heating rate, including a cooling run from 150 to −80 °C at 10 °C/min cooling rate, were carried out under nitrogen atmosphere (flow rate: 50 mL/min). The thermal events of the DSC traces of the TPUs (glass transition temperatures, melting temperature and cold crystallization) were obtained from the second DSC heating run.

#### 2.3.3. Thermal Gravimetric Analysis (TGA)

The structure of the TPUs was also assessed by TGA in a TGA Q500 equipment (TA Instruments, New Castle, DE, USA) under nitrogen atmosphere (flow rate: 50 mL/min). Each TPU sample (9–10 mg) was placed in a platinum crucible and heated from 35 to 800 °C by using a heating rate of 10 °C/min.

#### 2.3.4. X-ray Diffraction (XRD)

The crystallinity of the TPUs was assessed in a Bruker D8-Advance diffractometer (Bruker, Ettlingen, Germany); KαCu radiation was used.

#### 2.3.5. Plate-Plate Rheology

The viscoelastic properties of the TPUs were assessed by plate-plate rheology in a DHR-2 rheometer (TA Instruments, New Castle, DE, USA). TPU samples were placed and melted on the lower stainless steel plate and a stainless steel upper plate of 20 mm diameter was used. The gap used was 0.40 mm. Two kind of rheological tests were carried out:Temperature sweep tests. TPUs were melted at 120 °C and then cooled down from 120 to −20 °C by using a cooling rate of 5 °C/min. The frequency was 1 Hz.Frequency sweep tests. TPUs were placed between the two plates at 25 °C for 10 min and then the angular frequency was varied from 0.01 to 100 rad/s by using a strain amplitude of 2.5%.

#### 2.3.6. Probe Tack

Probe tack measurements of the TPU PSAs were performed in a TA.XT2i Texture Analyzer (Stable Micro Systems, Surrey, UK); a flat end cylindrical stainless-steel probe of 3 mm diameter was used. The TPU PSAs were prepared by placing 10 mL TPU solution made with 2 g solid TPU and 5 mL methyl ethyl ketone (MEK, Jaber Industrias Químicas, Madrid, Spain) on square stainless steel 304 plates with dimensions 60 × 60 × 1 mm^3^, and the thickness was adjusted with a metering rod of 400 µm. After MEK evaporation at room temperature for 24 h, TPU PSAs 30–50 µm thick were obtained. The probe tack test was carried out at 25 °C by applying 5 N force for 1 s (loading speed was 0.1 mm/s) on the TPU PSA surface and then pulled out at 10 mm/s, and a stress–strain curve was obtained. The maximum of the stress–strain curve was taken as the tack of the TPU PSA. At least five replicates were measured and averaged.

#### 2.3.7. 180° Peel Test

TPU PSAs were prepared by placing 10 mL TPU solution made with 2 g solid TPU and 5 mL MEK on a 50 µm-thick polyethylene terephthalate (PET) film; before applying the TPU solution, the PET film was wiped with MEK. The TPU solution was applied on the PET film with a pipette and spread by means of a metering rod of 400 μm. Then, MEK solvent was evaporated at room temperature for 24 h to obtain a TPU film of 30–40 µm thick on the PET film. The TPU PSA on PET samples were cut in 30 mm and 200 mm rectangles and joined to MEK wiped-aluminum 5754 plates of dimensions 30 mm × 150 mm × 1 mm by passing a 2 Kg rubber roller 30 times over the joint. After 1 h, the 180° peel tests of the aluminum 5754/TPU/polyethylene terephthalate (PET) joints (Figure 2) were carried out in an Instron 4411 universal testing machine (Instron Ldt., Buckinghamshire, UK) by using a pulling rate of 152 mm/min. At least five replicates were measured and averaged.

## 3. Results and Discussion

The main objective of this study is the design of TPU PSAs by changing the composition of the mixed polyether + polyester soft segments for changing their viscoelastic properties and for optimizing their adhesion and cohesion properties. The TPUs synthesized with mixed polyether soft segments showed good tack and good adhesion, but poor cohesion [37,38,39,40,41], whereas the ones made with polyester soft segments showed poor tack but high cohesion [2]. Therefore, the use of polyether + polyester blends should produce TPUs with optimal tack, adhesion and cohesion properties. Thus, in the first part of this section, the characterization of the TPUs synthesized with different PPG+PAd blends by using an NCO/OH ratio of 1.30 and the TPU PSAs obtained by placing thin TPU films on PET substrate will be shown. In the second part of this section, the NCO/OH ratio was varied between 1.13 and 1.30 by using 50 wt% PPG + 50 wt% PAd blend, and the TPUs and TPU PSAs obtained will be characterized. All TPUs have a low HS content (13–15%).

### 3.1. TPUs and TPU PSAs Synthesized with PPG+PAd Mixed Soft Segments by Using an NCO/OH Ratio of 1.30

In this study, ATR-IR spectroscopy and DSC were mainly used for assessing the degree of micro-phase separation and miscibility of the TPUs synthesized with PPG+PAd mixed SS. Several previous studies [19,20,21,22,23,24,25] have demonstrated that the degree of micro-phase separation of the TPUs can be assessed by using these experimental techniques. The ATR-IR spectrum of 1.30/PPG in Figure 3 shows the bands of the urethane HS at 3303 cm^−1^ (N-H stretching), 1728 cm^−1^ (C=O stretching), 1598 cm^−1^ (benzene ring of MDI) and 1535 cm^−1^ (N-H bending), and the ones of the PPG SS appear at 2941 and 2867 cm^−1^ (asymmetric and symmetric C-H stretching respectively) and 1091 cm^−1^ (C-O stretching of ether). The ATR-IR spectrum of 1.30/PAd shows the bands of the urethane HS at 3336 cm^−1^ (N-H stretching), 1724 cm^−1^ (C=O stretching), 1598 cm^−1^ (benzene ring of MDI) and 1531 cm^−1^ (N-H bending), and the ones of the PAd SS at 2953 and 2873 cm^−1^ (asymmetric and symmetric C-H stretching respectively) and 1168 cm^−1^ (C-O stretching of ester) (Figure 3). The ATR-IR spectra of the TPUs synthesized with PPG+PAd mixed soft segments show the absorption bands of 1.30/PPG and 1.30/Pad, but their relative intensities vary depending on the amounts of PPG and Pad (Figure 3). Thus, the urethane HS of the TPUs synthesized with PPG+PAd blends can be distinguished at 3309–3335 cm^−1^ (N-H stretching), 1725–1727 cm^−1^ (C=O stretching) and 1531–1534 cm^−1^ (N-H bending), and the SS appear at 2937–2970 cm^−1^ and 2868–2872 cm^−1^ (asymmetric and symmetric C-H stretching respectively) and 1095–1164 cm^−1^ (C-O stretching of ester and ether). The ATR-IR spectra of the TPUs made with PPG+PAd blends (Figure 3) show differences in the structure of the SS—C-H stretching, C-O stretching—and the HS—C=O stretching, N-H bending.

The degree of micro-phase separation of the TPUs can be estimated by curve fitting of the carbonyl stretching region at 1800–1650 cm^−1^ of the ATR-IR spectra. The assignment of the different C=O species was carried out according to previous studies [18,23,47]. Two contributions of free carbonyl (1728 cm^−1^) and associated by hydrogen bond (H-bonded) carbonyl (1708 cm^−1^) species can be distinguished in 1.30/PPG (Figure 4), and five C=O species can be differentiated in 1.30/PAd and all TPUs made with PPG+PAd mixtures (Figure 4 and Figure A1). The C=O species in the TPUs synthesized with PPG+PAd blends correspond to free C=O of ester (1739 cm^−1^), free C=O of urethane (1729–1730 cm^−1^), H-bonded C=O in disordered urethane (1721 cm^−1^), H-bonded C=O in ordered urethane (1708 cm^−1^) and H-bonded C=O in disordered soft domains (1688–1689 cm^−1^) (Figure 4 and Figure A1).

The percentages of the different C=O species in the TPUs are given in Table 2. The free carbonyl species are dominant in 1.30/PPG (TPU with higher micro-phase separation), and the H-bonded carbonyl species are dominant in 1.30/PAd (TPU with lower degree of micro-phase separation) (Table 2). The percentages of the free C=O species in the TPUs synthesized with PPG+PAd blends is similar (28–30%) and significantly lower than in 1.30/PPG, but the percentages of the free C=O ester species increase by increasing the PAd content in the TPUs made with polyols blends and, interestingly, they are higher than in 1.30/PAd. In general, the percentages of the H-bonded C=O in ordered hard domains and H-bonded C=O in disordered soft segments species are lower in the TPUs synthesized with PPG+PAd blends than in 1.30/Pad and the one of H-bonded C=O in disordered hard domains species is higher (Table 2). Furthermore, the structure of the hard domains is somewhat different in 1.30/PPG(25)PAd(75) than in the other TPUs synthesized with PPG+PAd blends because its lower percentages of the H-bonded C=O in disordered hard domains and H-bonded C=O in disordered SS than in 1.30/PAd (Table 2); this anticipates a different miscibility of the mixed SS.

Shen et al. [48] and Li et al. [49] have proposed a simple procedure for determining the mass fraction of the hard and soft segments, and the degree of micro-mixing of the hard and soft phases in the polyurethanes. This procedure was used in this study to verify the changes of the micro-mixing of the hard and soft phases in the TPUs made with mixed polyester and polyether SS. According to references [48,49], the ratio between the curve fitted C=O free carbonyl group and H-bonded carbonyl group bands can be used to calculate the fraction of hydrogen bonded urethane carbonyl groups, X_b_. In this study, the X_b_ value was calculated according to Equation (1):(1)$$X_{b}=\frac{A_{1723-1722}+A_{1708}}{{\;K^{\prime }A}_{1731-1729}+A_{1723-1722}+A_{1708}}$$

The values of X_b_ of the TPUs were calculated from the areas of A_H-bonded carbonyl_ and A_free carbonyl_ obtained from the integration of the curve fitted bands, a K’ value of 1.2 was used [48,49], and they increase by increasing the PAd content in the TPU (Table 3).

The mass fraction (W_2_) of the hard segments containing soft phases in the TPU was calculated according to Equation (2):(2)$$X_{b}=\frac{\left(1-X_{b}\right)f}{\left(1-X_{b}\right)f+\left(1-f\right)}$$
where f is the weight fraction of the hard segments.

The mass fraction of the TPUs in a state of micro-mixing (W_2_f), the mass fraction of the soft phase (SP) and the mass fraction of the hard phase (HP) were calculated according to reference [49]. Table 3 shows the X_b_, W_2_, W_2_f, SP and HP values for the TPUs. The mass fractions of SP and HP are similar in all TPUs and they agree with the values given in Table 1. However, the mass fraction of the hard segments containing soft phases and the micro-mixing of the hard and soft phases in the TPUs decrease steadily from 1.30/PPG to 1.30/PAd, the values of 1.30/PPG(75)PAd(25) are somewhat similar than in 1.30/PPG(50)PAd(50); this indicates different extent of miscibility between the soft segments.

The phase separation in polymer blends has also been evidenced by the existence of several glass transition temperatures in the DSC traces [22]; in this study, the miscibility of the SS in the TPUs was analyzed by DSC. Figure 5 shows the DSC traces of 1.30/PPG and 1.30/PAd which exhibit one glass transition temperature of the soft segments (T_g,s_) at −50 °C and −46 °C respectively (Table 4); on the other hand, 1.30/PPG is amorphous whereas 1.30/PAd shows soft crystalline domains (cold crystallization of the soft segments at −1 °C and melting at 43 °C). The DSC traces of all TPUs synthesized with PPG+PAd show soft crystalline domains and two T_g_s; this indicates a certain incompatibility between the PPG and PAd soft segments (Figure 5, Table 4). Thus, the T_g,s1_ and T_g,s2_ values decrease in 1.30/PPG(75)PAd(25) with respect to the parent TPUs, and the T_g,s1_ values decrease and the ones of T_g,s2_ slightly increase in 1.30/PPG(50)PAd(50) and 1.30/PPG(25)PAd(75). This in an indication of a different degree of miscibility in the TPUs depending on their PAd content, in agreement with previous studies [19,20,22,23,48,49]. Thus, the different degree of micro-phase separation in the TPUs with mixed soft segments is caused by the incompatibility of the PPG and PAd soft segments and the one induced by the segregation of the hard and soft segments [19,20,23]. On the other hand, the cold crystallization temperature in the TPUs with mixed soft segments increases in 1.30/PPG(50)PAd(50) and 1.30/PPG(25)PAd(75) and the melting temperature of the soft domains slightly decreases. Furthermore, the enthalpies of the cold crystallization and the melting of the SS increase by increasing the PAd content in the TPUs synthesized with PPG+PAd.

The evidence obtained from the DSC traces of the TPUs synthesized with PPG+PAd agrees well with their X-ray diffractograms (Figure 6). The X-ray difractogram of 1.30/PPG(75)PAd(25) shows a broad peak with a small crystalline peak at 2θ value of 21° indicating a dominant amorphous structure typical of the MDI+BD hard segments [50]. However, 1.30/PPG(50)PAd(50) and 1.30/PPG(25)PAd(75) show sharp peaks at 2θ values of 21, 22 and 24°, and these diffraction peaks correspond to the PAd soft segments (Figure A2). Furthermore, the intensities of the diffraction peaks are higher and the amorphous halo is less marked in 1.30/PPG(25)PAd(75), indicating higher crystallinity. Therefore, the addition of more or less than 50 wt% PAd changes differently the miscibility between the PPG and PAd soft segments and the crystallinity of the TPUs made with mixed soft segments, and this causes differences in their degree of micro-phase separation.

TGA has been used for assessing the structural features of the polyurethanes [37]. The TGA curves show that 1.30/PAd is more thermally stable than 1.30/PPG and 1.30/PPG(75)PAd(25) (Figure 7) and, therefore, the T_5%_ values (temperatures at which 5 wt% is lost) are higher (Table 5). The thermal stabilities of the TPUs synthesized with 50 wt% or more PAd are intermediate (283–292 °C) and they increase by increasing the PAd content. The derivative of the weight loss curves show the existence of two thermal degradations in the TPUs (Figure A3) due to the hard domains (298–329 °C) and the soft domains (365–395 °C) [38,39] (Table 5). The weight losses due to the hard and soft domains in the TPUs synthesized with PPG+PAd are somewhat similar because their similar HS content (Table 1), but the weight losses of the hard domains are higher than in 1.30/PPG and 1.30/Pad; this confirms the different miscibility and micro-mixing of the hard and soft phases in the TPUs made with mixed soft segments. On the other hand, the temperatures of decomposition of the hard domains are similar in 1.30/PPG(75)PAd(25) and 1.30/PPG(50)PAd(50), but the ones of the soft domains are higher in 1.30/PPG(50)PAd(50) because of its higher crystallinity.

The viscoelastic properties of the TPUs determine the properties of the PSAs and they are influenced by their degree of micro-phase separation [37,40]. Figure 8 shows the variation of the storage moduli (G′) of the TPUs as a function of the temperature. Considering that all TPUs have the same NCO/OH ratio, their viscoelastic properties should be mainly determined by the miscibility of the mixed soft segments and the degree of micro-mixing of the hard and soft phases. 1.30/PAd shows higher storage moduli than 1.30/PPG, and the storage moduli of 1.30/PAd decreases less steeply by increasing the temperature, this is due to the dominant contribution of the H-bonded C=O species. The storage moduli of 1.30/PPG(75)PAd(25) are intermediate among the ones of 1.30/PPG and 1.30/PAd, but unexpectedly the ones of 1.30/PPG(50)PAd(50) and 1.30/PPG(25)PAd(75) are lower than in 1.30/PPG and a more noticeable decrease of G′ by increasing the temperature is noticed, in agreement with their higher crystallinity and greater miscibility of the soft domains. The storage and loss moduli cross at a given temperature (a typical example is shown in Figure A4), i.e., at a temperature lower than the temperature of the cross-over between the storage and loss moduli (T_cross-over_) the elastic behavior is dominant in the TPU, whereas above T_cross-over_ the viscous behavior is dominant. All TPUs show a cross-over between the storage and loss moduli (Table 6), the value of T_cross-over_ is substantially lower in 1.30/PPG than in 1.30/PAd because of the absence of crystallinity in 1.30/PPG. Due to the miscibility between the PPG and PAd soft segments, the moduli at the cross-over (G_cross-over_) of the TPUs synthesized with PPG+PAd are higher than in 1.30/PPG and 1.30/PAd, and both the G_cross-over_ and T_cross-over_ values increase by increasing their PAd content. Furthermore, the values of T_cross-over_ of 1.30/PPG(75)PAd(25) and 1.30/PPG(50)PAd(50) are lower than the ones of 1.30/PPG and 1.30/PAd because the different degree of micro-mixing of the hard and soft phases caused by the incompatibility of the PPG and PAd soft segments and the one induced by the segregation of the hard and soft segments, in agreement with the ATR-IR, DSC and TGA results.

The viscoelastic properties of the pressure sensitive adhesives (PSAs) are tightly related to their adhesion properties, i.e., tack, shear resistance and peel strength [51]. Dahlquist established a relationship between the viscoelastic and the adhesion properties of the PSAs, it has been established that they should show tack when the storage moduli at 25 °C and 1 Hz are lower than 300 kPa [52]. Therefore, all TPUs should show tack (G′ values at 25 °C range between 5.5 and 256.9 kPa), except 1.30/PAd (Table 6), and the TPUs synthesized with PPG+PAd should have PSA properties.

Mazzeo proposed the use of the frequency sweep rheological tests for predicting the tack, the peel strength and the cohesion of the PSAs [53]. Thus, the tack was associated to low tan delta and storage moduli values, the cohesion/shear resistance was associated to high storage moduli values at low frequencies, and the peel strength was related to high loss moduli values at high frequencies [53]. On the other hand, Mazzeo’s study related the bonding of the PSAs to their elastic moduli values at low frequency and the debonding of the PSAs to the ratio of the elastic moduli at high and low frequencies. Similarly, Chu [54] related the bonding of the PSAs to the values of the storage moduli at a frequency of 0.01 rad/s, while the debonding of the PSAs was associated to the values of the storage and loss moduli at a frequency of 100 rad/s.

The variation of the storage (G′) and loss (G″) moduli of the TPUs at 25 °C as a function of the frequency obtained by oscillatory frequency sweeps rheological experiments is given in Figure 9. Similar findings than in the G′ vs. temperature plots (Figure 8) are obtained, the highest moduli correspond to 1.30/PPG(50)PAd(50) and 1.30/PPG(25)PAd(75). The storage moduli at a frequency of 0.01 rad/s in 1.30/PPG and 1.30/PPG(75)PAd(25) are lower than in 1.30/PPG(50)PAd(50) and 1.30/PPG(25)PAd(75) which have higher crystallinity and better miscibility of the soft segments, so high tack and low shear resistance are expected. On the other hand, all TPUs have high G″ values at a frequency of 100 rad/s so they should show good peel strength; however, good peel strength also requires high tan delta values at 100 rad/s [53], but the TPUs in this study have tan delta values lower than 1.5 (Table 6). Furthermore, the adequate debonding of the PSA requires G′(100 rad/s)/G′ (0.01 rad/s) values between 5 and 300 Pa (Table 7), and all TPUs are within this range except 1.30/PPG(25)PAd(75). Therefore, according to the viscoelastic properties, the best balance between tack, shear and peel properties in the TPUs made with mixed soft segments should correspond to 1.30/PPG(50)PAd(50) (Table 7).

The predicted behavior from the frequency sweep plate-plate rheological experiments of the TPUs as PSAs was compared with the experimental measurements of the probe tack and 180° peel strength of aluminum 5754/TPU PSA joints (Table 8). In general, a good agreement between the predicted and experimental tack and 180° peel strength is obtained, the highest tack and 180° peel strength correspond to 1.30/PPG and 1.30/PPG(75)PAd(25), and the highest cohesion belongs to 1.30/PPG(25)PAd(75)—it is the only TPU PSA showing an adhesion failure (Figure 10). The tack values of the TPU PSAs are within the range of other polyurethane PSAs [32,37,41,46], and the 180° peel strength values are similar to removable polyurethane PSAs, except in the joint made with 1.30/PPG which corresponds to high-shear polyurethane PSAs [37,38,40,41].

In summary, the addition of less than 50 wt% PAd produces moderate miscibility between the PPG and PAd soft segments and the crystallinity is almost absent; this leads to low moduli of the TPUs, which are related to high tack, 180° peel strength values and low cohesion. On the contrary, 1.30/PPG(50)PAd(50) and 1.30/PPG(75)PAd(25) show high miscibility of the PPG and PAd soft segments and high crystallinity, and their moduli are high and the cohesion is high; however, low tack and 180° peel strength values are obtained because the reduced mobility of the polymer chains.

### 3.2. TPUs and TPU PSAs Synthesized with 50 wt% PPG + 50 wt% PAd Soft Segments and Different NCO/OH Ratios

The miscibility between the PPG and PAd soft segments should also be affected by the NCO/OH ratio used in the synthesis of the TPUs. In this section, the structural and viscoelastic properties of the TPUs synthesized with 50 wt% PPG + 50 wt% PAd and NCO/OH ratios of 1.13, 1.20 and 1.30 were analyzed, and their potential as TPU PSAs has also been considered. The increase of the NCO/OH ratio slightly increases the HS content of the TPUs (from 13 to 15%—Table 1).

The degree of micro-phase separation of the TPUs synthesized with different NCO/OH ratios was assessed by curve fitting of the carbonyl region (1800–1650 cm^−1^) of the ATR-IR spectra and by DSC.

The carbonyl regions (1800–1650 cm^−1^) of the ATR-IR spectra of the TPUs are shown in Figure A5 and the percentages of C=O species obtained from the curve fittings (Figure A6) are given in Table 9. Small differences can be distinguished in the TPUs synthesized with different NCO/OH ratios, but the increase of the NCO/OH ratio increases the percentages of the H-bonded C=O in hard and soft domains species at the expense of the free C=O species. Thus, the increase of the NCO/OH ratio causes better miscibility of the PPG and PAd soft segments in the TPUs, in agreement with a previous study [19]. On the other hand, the micro-mixing of the hard and soft phases in the TPUs was estimated from the curve fitting of the carbonyl region of the ATR-IR spectra [48,49]. Table 10 shows the X_b_, W_2_, W_2_f, SP and HP values for the TPUs synthesized with different NCO/OH ratios. The mass fractions of SP and HP in the TPUs change similarly to the values given in Table 1, whereas the mass fraction of the hard segments containing soft phases increases slightly by increasing the NCO/OH ratio, and the micro-mixing of the hard and soft phases in the TPUs increases by increasing the NCO/OH ratio; however, 1.20/PPG(50)PAd(50) shows a lower W_2_f value than the other TPUs, which indicates a higher degree of micro-phase separation than in the other TPUs.

The DSC traces of all TPUs synthesized with different NCO/OH ratios show the glass transitions, the cold crystallization and the melting of the soft segments (Figure 11). Two glass transitions of the soft segments can be distinguished. The T_g,s1_ value decreases by increasing the NCO/OH ratio, anticipating better miscibility. The differences between the T_g,s1_ and T_g,s2_ values become smaller by increasing the NCO/OH ratio, although the T_g,s2_ value is somewhat lower in 1.20/PPG(50)PAd(50) (Table 11). On the other hand, the temperatures and enthapies of the cool crystallization and melting of 1.13/PPG(50)PAd(50) and 1.30/PPG(50)PAd(50) are somewhat similar, but the temperatures of the cool crystallization and melting are lower and their enthalpies are higher in 1.20/PPG(50)/PAd(50) (Table 11). Therefore, the interactions between the soft segments are more net in 1.20/PPG(50)PAd(50) than in the other TPUs.

Figure 12 shows the X-ray diffractograms of the TPUs synthesized with different NCO/OH ratios. All TPUs show a wide amorphous peak over which three sharp diffraction peaks at 2θ values of 21°, 22° and 24° corresponding to the PAd soft segments crystallites can be distinguished. The amorphous phase in 1.20/PPG(50)/PAd(50) is less than in 1.30/PPG(50)/PAd(50) and 1.13/PPG(50)/PAd(50), which indicates more net interactions between the soft segments, in agreement with the DSC experiments.

Figure 13 shows the variation of the storage (G′) and loss (G″) moduli as a function of temperature for TPUs synthesized with different NCO/OH ratios. The moduli of the TPUs increase by increasing the NCO/OH ratio because of the slight increase of the HS content. However, at low and high frequencies, the moduli of 1.20/PPG(50)PAd(50) and 1.30/PPG(50)PAd(50) are somewhat similar due to the more net interactions between the soft segments in 1.20/PPG(50)PAd(50). Furthermore, the decrease of the moduli of the TPUs by decreasing the frequency is made less steep by increasing the NCO/OH ratio, likely due to the slight increase of the HS content. 1.13/PPG(50)PAd(50) has low storage moduli at a frequency of 0.01 rad/s, so high tack but low shear resistance is expected; on the contrary, 1.20/PPG(50)PAd(50) and 1.30/PPG(50)PAd(50) with better miscibility of the SS should have high shear resistance (they have high G’ values at a frequency of 0.01 rad/s) but low tack. On the other hand, all TPUs have high G″ values at a frequency of 100 rad/s so they should show good peel strength. Furthermore, the adequate debonding of the PSA requires G′(100 rad/s)/G′ (0.01 rad/s) values between 5 and 300 Pa, all TPUs are within this range (269 for 1.13/PPG(50)PAd(50), 301 for 1.20/PPG(50)PAd(50), 62 for 1.30/PPG(50)PAd(50)). Therefore, according to the viscoelastic properties, the best balance between tack, shear and peel properties in the TPUs should correspond to 1.20/PPG(50)PAd(50).

The predicted behavior from the frequency sweep plate-plate rheological experiments of the TPUs was compared with the experimental measurements of the probe tack and 180° peel strength of aluminum 5754/TPU PSA joints (Table 12). In general, a good agreement between the predicted and experimental tack is obtained; the highest tack corresponds to 1.20/PPG(50)PAd(50). However, the 180° peel strength values are very low because the miscibility of the PPG and PAd soft segments inhibits the mobility of the polymer chains. Furthermore, the cohesion of the TPU PSAs is poor, likely due to the low HS content. Therefore, the TPUs synthesized with PPG+PAd soft segments made with different NCO/OH ratios are promising PSAs, but their cohesion needs to be increased.

## 4. Conclusions

The structure, viscoelastic and pressure sensitive adhesion properties of the TPUs synthesized with PPG and PAd mixed soft segments and low hard segments content (13–15%) and different NCO/OH ratios were investigated. The increase of the PAd content diminished the miscibility of the mixed soft segments and increased the crystallinity of the TPUs. The different micro-mixing of the hard and soft phases in the TPUs with mixed soft segments was caused by the incompatibility of the PPG and PAd soft segments and the segregation of the hard and soft segments.

The percentages of the free C=O species in the TPUs synthesized with PPG+PAd blends were similar (28–30%) and significantly lower than in 1.30/PPG. In general, the percentages of the H-bonded C=O in ordered hard domains and H-bonded C=O in disordered soft segments species were lower and the one of H-bonded C=O in disordered hard domains species was higher in the TPUs synthesized with PPG+PAd blends than in 1.30/Pad. The structure of the hard domains was somewhat different in 1.30/PPG(25)PAd(75) than in the other TPUs synthesized with mixed soft segments because of its dominant amorphous structure. On the other hand, the TPUs synthesized with mixed soft segments showed soft crystalline domains, and the values of T_g,s1_ and T_g,s2_ decreased in 1.30/PPG(75)PAd(25) with respect to the parent TPUs. 1.30/PPG(50)PAd(50) and 1.30/PPG(25)PAd(75) showed higher cold crystallization temperature and sharp peaks at 2θ values of 21, 22 and 24° in the X-ray diffractograms.

The viscoelastic properties of the TPUs made with mixed soft segments were mainly determined by the miscibility of the mixed soft segments and the storage moduli of 1.30/PPG(75)PAd(25) were intermediate among the ones of the parent TPUs, but the ones of 1.30/PPG(50)PAd(50) and 1.30/PPG(25)PAd(75) were lower than in 1.30/PPG, in agreement with their higher crystallinity and greater miscibility of the soft domains. All TPUs synthesized with PPG+PAd showed a cross-over between the storage and loss moduli, and the moduli at the cross-over were higher than in the parents TPUs, due to the miscibility between the PPG and PAd soft segments; both the moduli and the temperatures at the cross-over increased by increasing their PAd content.

The increase of the NCO/OH ratio increased the percentages of the H-bonded C=O in hard and soft domains species at the expense of the free C=O species in the TPUs and improved the miscibility of the PPG and PAd soft segments. The temperatures and enthapies of the cool crystallization and melting of 1.13/PPG(50)PAd(50) and 1.30/PPG(50)PAd(50) were somewhat similar, but the temperatures of the cool crystallization and melting were lower and their enthalpies were higher in 1.20/PPG(50)/PAd(50). The amorphous phase in 1.20/PPG(50)/PAd(50) is less important than in the other TPUs; this indicated more net interactions between the soft segments. The moduli of the TPUs increased by increasing the NCO/OH ratio. 1.13/PPG(50)PAd(50) should have high tack and low shear resistance, and 1.20/PPG(50)PAd(50) and 1.30/PPG(50)PAd(50) should have high shear resistance but low tack, due to better miscibility of the soft segments.

All TPUs had high G″ values at a frequency of 100 rad/s so they should show good peel strength, and the G′(100 rad/s)/G′ (0.01 rad/s) values were between 5 and 300 Pa, indicating good debonding properties. In general, a good agreement between the predicted and experimental tack was obtained in the TPU PSAs, the highest tack corresponded to 1.30/PPG, 1.30/PPG(75)PAd(25) and 1.20/PPG(50)PAd(50), and the highest 180° peel strength values were found in 1.30/PPG and 1.30/PPG(75)PAd(25); however, lower 180° peel strength values were obtained in the TPUs made with different NCO/OH ratios. Furthermore, the cohesion of the TPU PSAs was insufficient, except in 1.30/PPG(25)PAd(75). Therefore, the TPUs synthesized with PPG + PAd soft segments were promising PSAs.

## Figures and Tables

**Figure 1 polymers-13-03097-f001:**
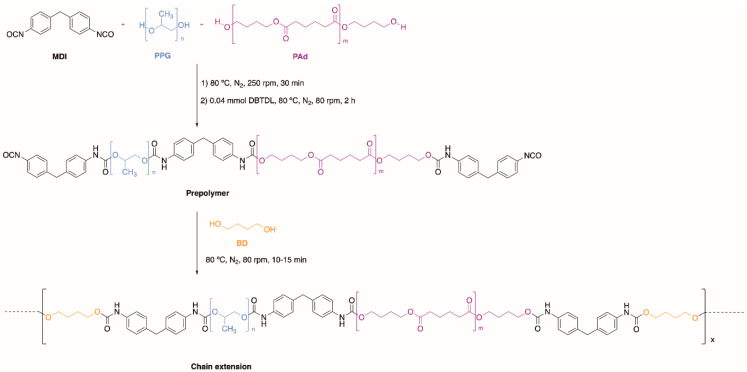
Scheme of the synthesis of the TPUs made with mixed PPG and PAd soft segments.

**Figure 2 polymers-13-03097-f002:**
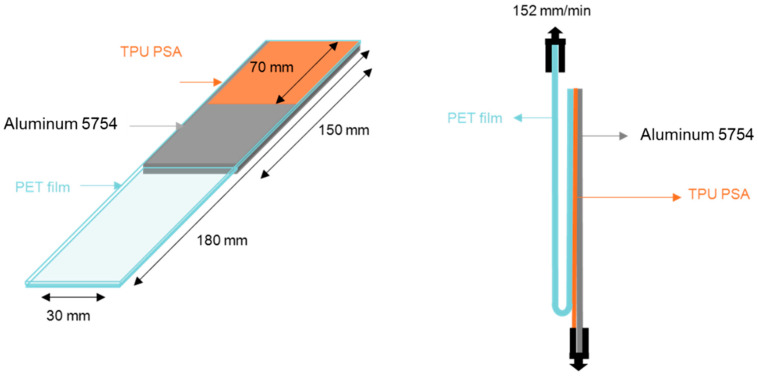
Scheme of the 180° peel test of aluminum 5754/TPU/polyethylene terephthalate (PET) joint.

**Figure 3 polymers-13-03097-f003:**
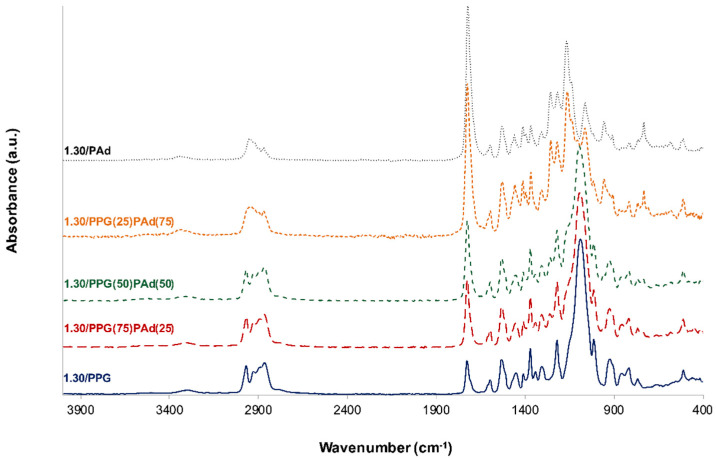
ATR-IR spectra of the TPUs synthesized with PPG, PAd and PPG+PAd blends.

**Figure 4 polymers-13-03097-f004:**
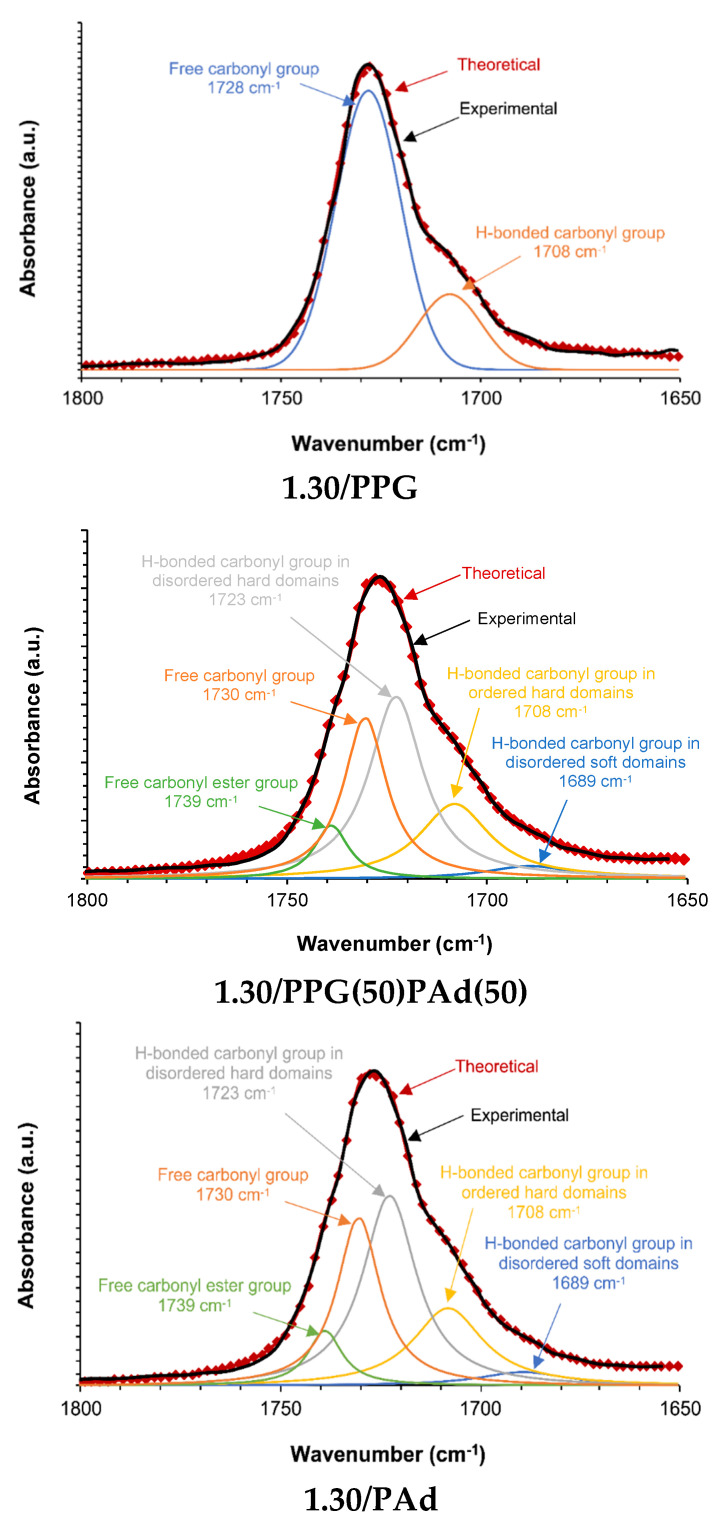
Curve fitting of the carbonyl region of the ATR-IR spectra of 1.30/PPG, 1.30/PPG(50)PAd(50) and 1.30/PAd. The term “theoretical” was used to describe the curve fitted spectra by using Gaussian function.

**Figure 5 polymers-13-03097-f005:**
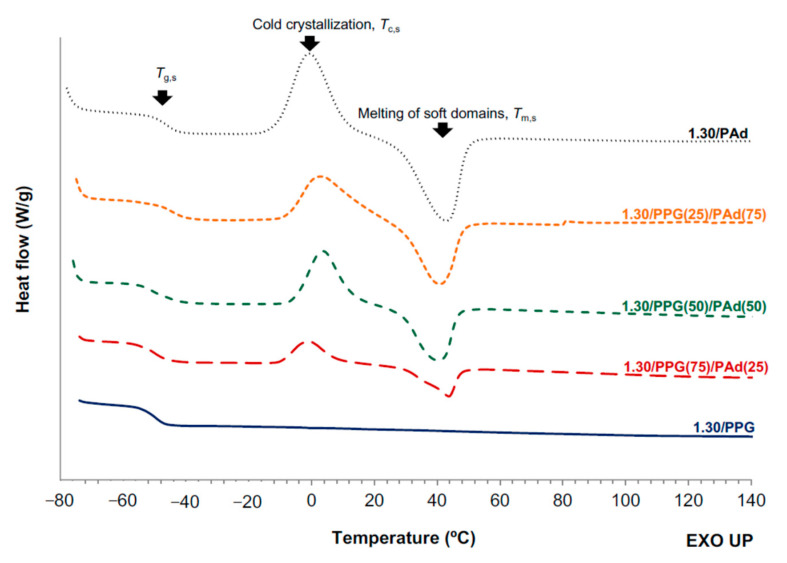
DSC traces of the TPUs synthesized with PPG, PAd, and PPG+PAd blends. Second heating run.

**Figure 6 polymers-13-03097-f006:**
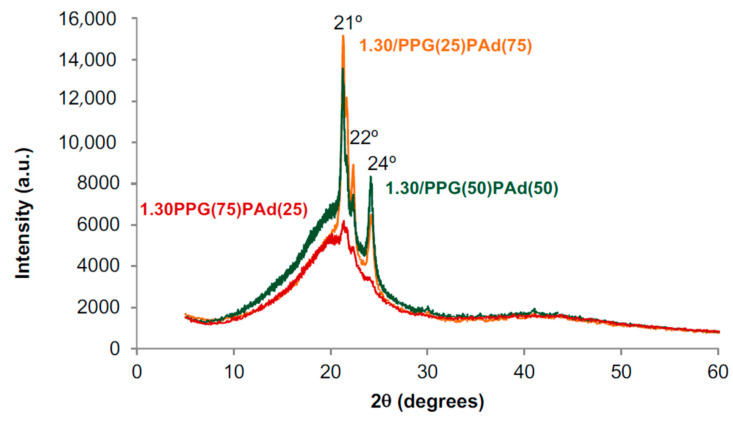
X-ray diffractograms of the TPUs synthesized with PPG+PAd blends.

**Figure 7 polymers-13-03097-f007:**
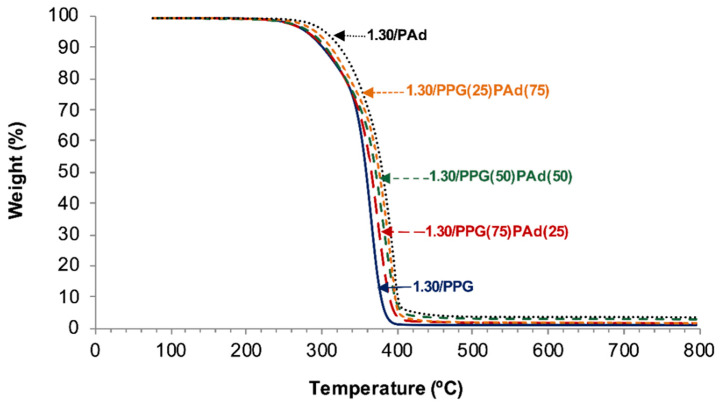
TGA curves of the TPUs synthesized with PPG, PAd, and PPG+PAd blends.

**Figure 8 polymers-13-03097-f008:**
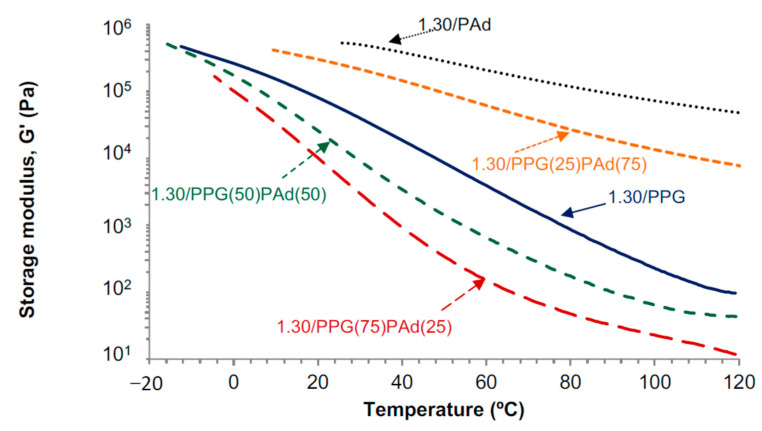
Variation of the storage modulus (G′) as a function of the temperature for the TPUs. Temperature sweep plate-plate rheology experiments.

**Figure 9 polymers-13-03097-f009:**
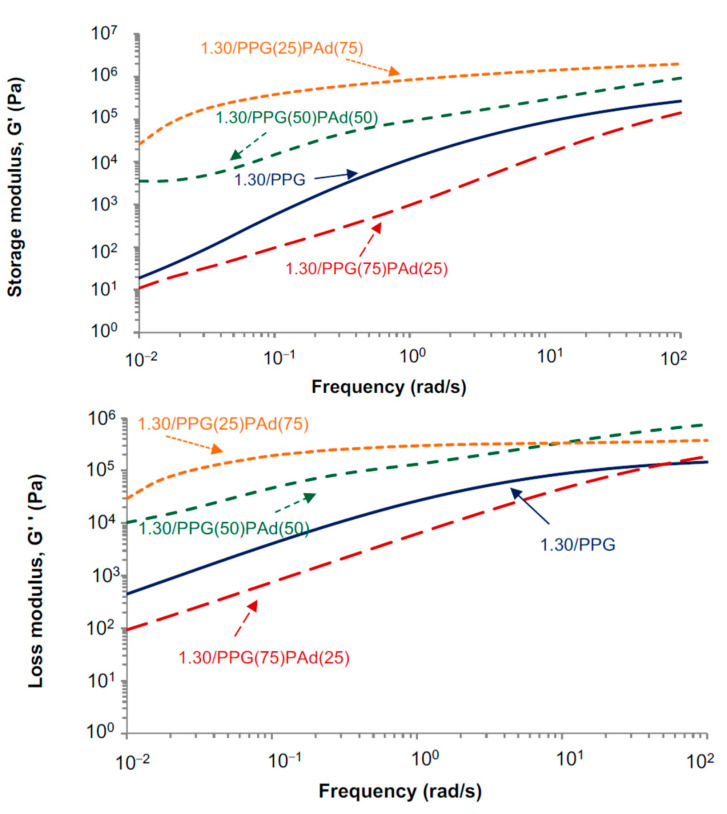
Variation of the storage (G′) and loss (G″) moduli at 25 °C as a function of the frequency for the TPUs. Frequency sweep plate-plate rheology experiments.

**Figure 10 polymers-13-03097-f010:**
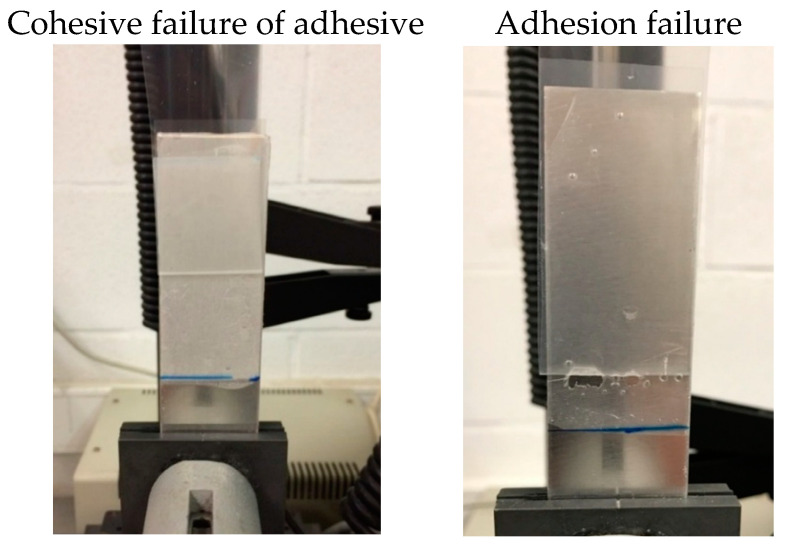
Loci of failure of aluminum 5754/TPU PSA joints.

**Figure 11 polymers-13-03097-f011:**
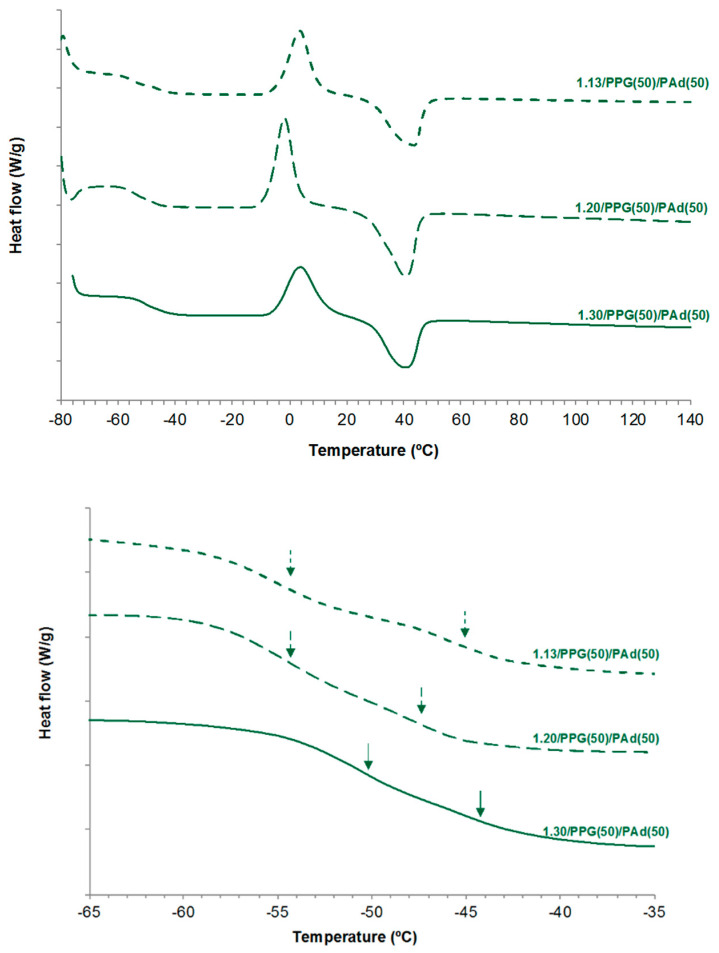
DSC traces of the TPUs synthesized with different NCO/OH ratios. Second heating run.

**Figure 12 polymers-13-03097-f012:**
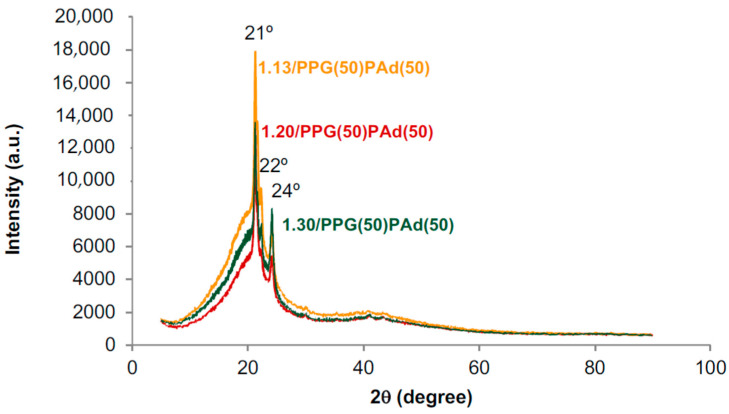
X-ray diffractograms of the TPUs synthesized with different NCO/OH ratios.

**Figure 13 polymers-13-03097-f013:**
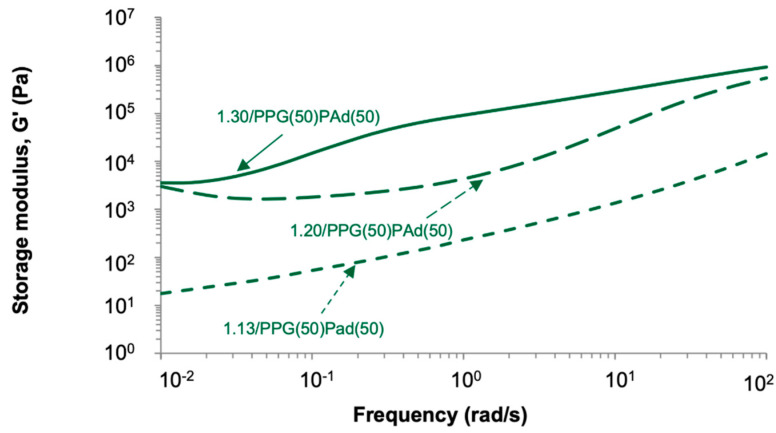
Variation of the storage (G′) and loss (G″) moduli as a function of the frequency of the TPUs synthesized with different NCO/OH ratios. Frequency sweep plate-plate rheology experiments.

**Table 1 polymers-13-03097-t001:** Nomenclatures and compositions of the TPUs.

Nomenclature	PPG (g)	PAd (g)	1,4-BD (g)	NCO/OH	HS (%) ^1^
1.30/PPG	80	-	1.08	1.30	15
1.30/PPG(75)PAd(25)	60	20	1.08	1.30	15
1.30/PPG(50)PAd(50)	40	40	1.08	1.30	15
1.30/PPG(25)PAd(75)	20	60	1.08	1.30	15
1.30/PAd	-	80	1.08	1.30	15
1.13/PPG(50)PAd(50)	40	40	0.47	1.13	13
1.20/PPG(50)PAd(50)	40	40	0.73	1.20	14

^1^ HS: Weight fraction of hard segments (HS) calculated as HS = 100 × [MDI weight + BD weight]/[total weight].

**Table 2 polymers-13-03097-t002:** Percentages of C=O species of the TPUs. Curve fitting of the carbonyl region of the ATR-IR spectra.

TPU	Free C=O Ester (%)—1739 cm^−1^	Free C=O (%)—1731–1728 cm^−1^	H-Bonded C=O in Disordered Hard Domains (%)—1723–1721 cm^−1^	H-Bonded C=O in Ordered Hard Domains (%)—1708 cm^−1^	H-Bonded C=O in Disordered Soft Domains (%)—1689–1688 cm^−1^
1.30/PPG	-	80	-	20	-
1.30/PPG(75)PAd(25)	7	30	42	20	1
1.30/PPG(50)PAd(50)	7	28	39	21	5
1.30/PPG(25)PAd(75)	5	28	31	24	2
1.30/PAd	2	24	36	32	5

**Table 3 polymers-13-03097-t003:** Main parameters for determining the micro-mixing of the hard and soft phases in the TPUs.

Nomenclature	X_b_	W_2_	W_2_f	SP	HP
1.30/PPG	0.172	0.1274	0.0191	0.87	0.13
1.30/PPG(75)PAd(25)	0.638	0.0600	0.0090	0.86	0.14
1.30/PPG(50)PAd(50)	0.643	0.0592	0.0089	0.86	0.14
1.30/PPG(25)PAd(75)	0.657	0.0570	0.0086	0.86	0.14
1.30/PAd	0.702	0.050	0.0075	0.86	0.14

**Table 4 polymers-13-03097-t004:** Thermal events from the DSC traces of the TPUs. Second heating run.

TPU	T_g,s1_ (°C)	T_g,s2_ (°C)	T_c_ (°C)	ΔH_c_ (J/g)	T_m,s_ (°C)	ΔH_m,s_ (J/g)
1.30/PPG	−50	--	--	--	--	--
1.30/PPG(75)PAd(25)	−55	−49	−1	8	44	9
1.30/PPG(50)PAd(50)	−51	−45	4	19	40	19
1.30/PPG(25)PAd(75)	−53	−44	3	23	41	22
1.30/PAd	--	−46	−1	34	43	34

**Table 5 polymers-13-03097-t005:** Temperatures and weight losses of the thermal decompositions of the TPUs. TGA experiments.

TPU	T_5%_ (°C) ^1^	T_1_ (°C)	Weight Loss_1_ (wt%)	T_2_ (°C)	Weight Loss_2_ (wt%)
1.30/PPG	280	298	17	365	82
1.30/PPG(75)PAd(25)	278	314	20	373	78
1.30/PPG(50)PAd(50)	283	313	20	381	77
1.30/PPG(25)PAd(75)	292	324	21	392	77
1.30/PAd	305	329	19	395	78

^1^ Temperature at which 5 wt% is lost.

**Table 6 polymers-13-03097-t006:** Temperature (T_cross-over_) and modulus (G_cross-over_) at the cross-over of the storage and loss moduli of the TPUs. Temperature sweep plate-plate rheology experiments.

TPU	T_cross-over_ (°C)	G_cross-over_ (Pa)	G′ at 25 °C and 1 Hz (kPa)
1.30/PPG	20	9.4·10^4^	57.6
1.30/PPG(75)PAd(25)	−4	1.6·10^5^	5.5
1.30/PPG(50)PAd(50)	1	1.5·10^5^	15.1
1.30/PPG(25)PAd(75)	52	1.1·10^5^	256.9
1.30/PAd	123	4.4·10^4^	529.1

**Table 7 polymers-13-03097-t007:** Viscoelastic properties of the TPUs related to their PSA characteristics. Frequency sweep plate-plate rheology experiments.

Rheological Data	1.30/PPG	1.30/PPG(75)PAd(25)	1.30/PPG(50)PAd(50)	1.30/PPG(25)PAd(75)
Tan delta (100 rad/s)	1.3	1.8	1.3	0.4
5 < G’(100 rad/s)/G’(0.1 rad/s) < 300	463	1443	62	5

**Table 8 polymers-13-03097-t008:** Probe tack and 180° peel strength values, and loci of failure of aluminum 5754/TPU PSA joints.

TPU PSA	Probe Tack (kPa)	180° Peel Strength (N/cm)	Locus of Failure
1.30/PPG	992 ± 26	2.4 ± 0.3	Cohesion of adhesive
1.30/PPG(75)PAd(25)	885 ± 9	0.6 ± 0.0	Cohesion of adhesive
1.30/PPG(50)PAd(50)	110 ± 7	0.2 ± 0.0	Cohesion of adhesive
1.30/PPG(25)PAd(75)	0.3 ± 0.0	0.2 ± 0.0	Adhesion

**Table 9 polymers-13-03097-t009:** Percentages of C=O species of the TPUs synthesized with different NCO/OH ratios. Curve fitting of the carbonyl region of the ATR-IR spectra.

TPU	Free C=O Ester (%)—1739 cm^−1^	Free C=O (%)—1731–1728 cm^−1^	H-Bonded C=O in Disordered Hard Domains (%)—1723–1721 cm^−1^	H-Bonded C=O in Ordered Hard Domains (%)—1708 cm^−1^	H-Bonded C=O in Disordered Soft Domains (%)—1689–1688 cm^−1^
1.13/PPG(50)PAd(50)	10	32	36	18	4
1.20/PPG(50)PAd(50)	9	30	37	20	4
1.30/PPG(50)PAd(50)	7	28	39	21	5

**Table 10 polymers-13-03097-t010:** Main parameters for determining the micro-mixing of the hard and soft phases in the TPUs.

Nomenclature	X_b_	W_2_	W_2_f	SP	HP
1.13/PPG(50)PAd(50)	0.5818	0.0588	0.0082	0.87	0.13
1.20/PPG(50)PAd(50)	0.6144	0.0591	0.0077	0.88	0.12
1.30/PPG(50)PAd(50)	0.6430	0.0592	0.0089	0.86	0.14

**Table 11 polymers-13-03097-t011:** Thermal events from the DSC traces of the TPUs synthesized with different NCO/OH ratios. Second heating run.

TPU	T_g,s1_ (°C)	T_g,s2_ (°C)	T_c_ (°C)	ΔH_c_ (J/g)	T_m,s_ (°C)	ΔH_m,s_ (J/g)
1.13/PPG(50)PAd(50)	−55	−46	3	19	43	20
1.20/PPG(50)PAd(50)	−54	−49	−2	21	41	24
1.30/PPG(50)PAd(50)	−51	−45	4	19	40	19

**Table 12 polymers-13-03097-t012:** Probe tack and 180° peel strength values strength of aluminum 5754/TPU PSA with different NCO/OH ratios joints, and loci of failure.

TPU PSA	Probe Tack (kPa)	180° Peel Strength (N/cm)	Locus of Failure
1.13/PPG(50)PAd(50)	126 ± 11	0.1 ± 0.0	Cohesion of adhesive
1.20/PPG(50)PAd(50)	251 ± 17	0.1 ± 0.0	Cohesion of adhesive
1.30/PPG(50)PAd(50)	110 ± 7	0.2 ± 0.0	Cohesion of adhesive

## Data Availability

The data presented in this study are available on request from the corresponding author.

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
