# Peer review of "Structural and Viscoelastic Properties of Thermoplastic Polyurethanes Containing Mixed Soft Segments with Potential Application as Pressure Sensitive Adhesives"

_polymers, 2021, doi:10.3390/polym13183097_

Round 1
Reviewer 1 Report
Please see my comments below and accordingly revise the manuscript
- The abstract is way too long. Concise it by putting only important information and key findings
- The novelty of the work should be better explained at the end of the introduction section
- Figure 1 should be more clearer
- Add in the experimental setup for the peel test, manufactured samples for testing
- Line 371-374, please elaborate
- The work and the explanations should be properly related to existing literature, which is missing on many occasions, please add-in
- Please add in the failure images of the peel tested samples, and some microscopic investigation if were carried out
- Conclusion is way too lengthy, please concise it with substance
Reviewer 2 Report
Manuscript ID: polymers-1349390
Title: Structural and viscoelastic properties of thermoplastic polyurethanes containing mixed soft segments with potential applications as pressure sensitive adhesives
The manuscript presents an experimental work focused on the development with blends of poly(propylene glycol) (PPG) and poly(1,4-butylene adipate) (PAd) polyols, di-phenylmethane-4,4’-diisocyanate (MDI) and 1,4-butanediol (BD) chain ex-tender to be used in the preparation of thermoplastic polyurethanes (TPUs). The overall results presented are good, and the data may be obtainable. I recommend publication of the work after the authors address my concern, which is indeed major.
This manuscript should be revised according to the following comments:
- The abstract is too long, the abstract should be used as a short narrative of the main point of the article, and the references in the article are too old, the recent references should be updated and cited.
- Line 211 shows that 1,4 BD is added to the prepolymer solution, but the added weight is not shown. Therefore, I would like to ask whether 1,4 BD will complete the reaction of all -NCO groups at the end? And whether the reaction process is monitored?
- What is the molecular weight of all polymers? Since molecular weight can greatly affect the physical properties of polymers, I hope to perform gel permeation chromatography to monitor the molecular weight of all samples.
- The 1.30/PPG has an obvious peak at 1689cm-1 in Figure 4, I don’t understand why 1.30/PAd shows it, but 1.30/PPG doesn’t. Curve fitting is performed in Figure 4 and Table 8, so all FTIR figures for curve fitting should be included in the article. In recent articles, Curve fitting has been used to calculate the mass fraction of the soft and hard phases of polyurethane, the author can calculate the mass fraction to further verify the change of polyurethane microphase separation (https://doi.org/10.1016/j.polymer.2019.05.010、https://doi.org/10.1016/j.porgcoat.2020.105702)
- How are High, Medium, Medium-Low, and Low defined in Table 6? the definition should be clearly explained in the article
- The author should compare other articles on the performance of polyurethane used in pressure-sensitive adhesives.
Overall, it is recommended for publication in the Polymers after the major revisions.
Reviewer 3 Report
The manuscript does not have any novelty. A large abstract, a large Introduction, a large Conclusions sections demonstrate that there is no important problem and its solution which should be outlined in the manuscript but many usual measurements in the area of TPU.
Line 68 says that the subject was "scarcely studied [17-21]" (5 references!) is not a ground for continuation of any research.
Possibly the manuscript will be reworked and resubmitted to another Journal. I pay attention to lined 178-179 (incoherent).
"Micro-phase separation" (lines 26, 187, others) should be experimentally proved or clearly stated that it is a plausible speculation.
Line 356, Fig. 4: "Theoretical curve". What theory are we talking about?
Line 129. "Da" should be replaced by g/mol.
Round 2
Reviewer 2 Report
The MS was properly revised according to the my suggestions. I recommend its publication without any further changes. But the revised manuscript not satisfied Reviewer #3 concerns. Please consider more deeply.
Reviewer 3 Report
The authors made cosmetic corrections. I have the same objections. In my opinion, the manuscript should be rejected.